# Cross-Subject EEG Emotion Recognition Using SSA-EMS Algorithm for Feature Extraction

**DOI:** 10.3390/e27090986

**Published:** 2025-09-21

**Authors:** Yuan Lu, Jingying Chen

**Affiliations:** 1Normal College, Jimei University, Xiamen 361021, China; ly21@jmu.edu.cn; 2National Engineering Research Center for E-Learning, Central China Normal University, Wuhan 430079, China

**Keywords:** EEG, emotion recognition, EMS, SSA, cross-subject

## Abstract

This study proposes a novel SSA-EMS framework that integrates Singular Spectrum Analysis (SSA) with Effect-Matched Spatial Filtering (EMS), combining the noise-reduction capability of SSA with the dynamic feature extraction advantages of EMS to optimize cross-subject EEG-based emotion feature extraction. Experiments were conducted using the SEED dataset under two evaluation paradigms: “cross-subject sample combination” and “subject-independent” assessment. Random Forest (RF) and SVM classifiers were employed to perform pairwise classification of three emotional states—positive, neutral, and negative. Results demonstrate that the SSA-EMS framework achieves RF classification accuracies exceeding 98% across the full frequency band, significantly outperforming single frequency bands. Notably, in the subject-independent evaluation, model accuracy remains above 96%, confirming the algorithm’s strong cross-subject generalization capability. Experimental results validate that the SSA-EMS framework effectively captures dynamic neural differences associated with emotions. Nevertheless, limitations in binary classification and the potential for multimodal extension remain important directions for future research.

## 1. Introduction

The electroencephalogram (EEG) holds significant value for emotion recognition due to their high temporal resolution and non-invasive nature [1,2]. However, their practical application faces numerous challenges stemming from the inherent biophysical properties of EEG signals and the complexity of emotions themselves [3,4]. EEG signals are characterized by inherent non-stationarity and a low signal-to-noise ratio, making them highly susceptible to interference from physiological artifacts [5,6]. Although many analytical methods rely on the assumption of stationarity for practicality, this property is particularly prominent during transient events like seizures or in ERP analyses [7]. The challenges become even more pronounced in cross-subject EEG emotion recognition tasks [8,9]. Beyond the intrinsic complexity of the signals, individual differences in physiology and psychology further exacerbate the difficulty of EEG-based emotion recognition [10,11]. As a result, EEG emotion recognition models often perform well in within-subject scenarios but demonstrate significantly reduced generalization capability in cross-subject settings [12,13].

To address this challenge, researchers have proposed various innovative methods to enhance the cross-subject adaptability of models by improving signal processing techniques or machine learning frameworks [14,15]. Anuragi A et al. proposed an automated cross-subject emotion recognition framework based on EEG signals, employing the Fourier-Bessel Series Expansion-based Empirical Wavelet Transform (FBSE-EWT) method. They demonstrated the effectiveness of the proposed framework using machine learning models such as Artificial Neural Networks (ANN) and k-Nearest Neighbors (k-NN), showing superior performance in human emotion classification compared to other state-of-the-art emotion classification models [16]. Wenhui Guo and Yaxuan Li et al. introduced a Functional Connectivity-Enhanced Feature Group Attention Network (FC-FAN) for cross-subject emotion recognition. Comprehensive evaluations indicate that this framework achieves outstanding performance in cross-subject emotion recognition tasks, effectively capturing the spatiotemporal features of EEG signals while clearly analyzing the nonlinear relationships among electrode signals [17]. Yong Peng and Honggang Liu et al. proposed a joint EEG feature transfer and semi-supervised cross-subject emotion recognition model, which achieves global optimality by simultaneously optimizing the shared subspace projection matrix and target labels. The joint learning paradigm of this model significantly improves emotion recognition performance on both the SEED (The SJTU Emotion EEG Dataset) and SEED-IV datasets [18].

Schurger et al. proposed EMS algorithm [19]. Traditional spatial filtering methods rely on predefined spatial regions or fixed spatial filters, whose static nature limits their ability to dynamically adapt to the temporal evolution of neural activity. The core advantage of EMS lies in its “experiment-effect-driven” characteristic, which constructs time-varying spatial filters to capture the spatiotemporal dynamics of emotional differences [19]. Although the EMS algorithm can effectively track the dynamic changes in experimental effects, its direct construction of filters based on raw signal differences makes it susceptible to interference from high-frequency noise and artifacts. To address this issue, this study proposes a novel framework, SSA-EMS, which integrates EMS with SSA. The SSA-EMS framework leverages the strengths of SSA in time series denoising and reconstruction, as well as the advantages of EMS in spatial filtering and dynamic adaptation, aiming to achieve effective feature extraction from EEG signals.

## 2. Dataset and Methods

### 2.1. Dataset

The SEED dataset, provided by Shanghai Jiao Tong University, is a publicly available dataset primarily used for EEG-based affective computing research [20]. This dataset contains EEG signals from 15 Chinese participants (7 males and 8 females). EEG signals were recorded using a 62-channel electrode cap based on the international 10–20 system, covering major brain regions. Each participant completed 15 trials, during which they watched 15 film clips, each approximately 4 min long. These clips were carefully selected and edited to effectively elicit one of three distinct target emotions: positive, neutral, or negative. Prior to each clip, a 5 s cue was presented, followed by a 45 s period for self-assessment after viewing. A 15 s rest period was provided between consecutive clips. The presentation order of the films was specifically designed to prevent two clips targeting the same emotion from being shown consecutively [20]. The entire experiment was conducted over three sessions per participant, with sessions spaced approximately one week apart. Each session corresponds to one data file, resulting in a total of 45 files. Data in each file are stored as a matrix in the format “channels × data points” and saved in .mat format. To protect privacy, participant identities are anonymized using numerical identifiers [20].

This study utilized the “Preprocessed_EEG” EEG data files provided by the SEED dataset, as these files have already undergone downsampling and preprocessing. The data were downsampled to 200 Hz and filtered using a 0–75 Hz bandpass filter [21].

### 2.2. Methods

#### 2.2.1. EMS Algorithm

Effect-Matched Spatial Filtering (EMS) is a data-driven, effect-oriented spatial filtering method designed to reduce high-dimensional EEG data into time series that reveal specific experimental effects. Grounded in matched filter theory and operating under the assumption of independent and identically distributed (i.i.d.) noise, EMS dynamically constructs spatial filters that align with the experimental effect of interest. At each time point, the method estimates, in a data-driven manner, a spatial filter that captures the spatial pattern of differences between experimental conditions. The original data are then projected onto this adaptively derived “functional axis” in sensor space [19]. Unlike methods based on anatomical landmarks or prior assumptions, EMS derives its filters directly from the empirically observed contrast patterns between experimental conditions. Furthermore, the filter weights are adjusted dynamically over time to maximize the expression of the targeted experimental effect in the sensor space, thereby enhancing the temporal resolution and sensitivity of effect detection.

The core mathematical principle of the EMS algorithm is to find the optimal projection direction ω that maximizes the difference between two classes of signals.$$\omega =\arg\max_{w}\frac{w^{T}(\mu_{1}-\mu_{0})}{\left\|w\right\|}$$

Here, $\mu_{1}$ and $\mu_{0}$ represent the average response patterns under two different conditions, respectively. The analytical solution to this optimization problem is given by:$$\omega =\frac{\mu_{1}-\mu_{0}}{\|\mu_{1}-\mu_{0}{\|}_{2}}.$$

The geometric interpretation is that by constructing the normal vector of a hyperplane, the signed distance from each data to the hyperplane is computed, where a positive value indicates that the data lies on the side pointed to by the normal vector, and a negative value indicates that it lies on the opposite side.

#### 2.2.2. SSA Algorithm

Singular Spectrum Analysis (SSA) is a non-parametric time series analysis method based on linear algebra and dynamical systems theory [22,23,24,25]. Its core idea is to decompose a complex time series into several components with clear physical interpretations (such as trend, periodicity, and noise) through a “decomposition–reconstruction” procedure, thereby achieving objectives such as denoising, forecasting, and periodicity extraction [26,27]. Compared with traditional methods such as Fourier transform and wavelet transform, SSA does not rely on the assumption of data stationarity, thus exhibiting greater adaptability, especially for analyzing non-stationary and nonlinear time series [28,29,30]. The SSA algorithm consists of two main stages—decomposition and reconstruction [31,32,33]—and can be specifically divided into the following four key steps:

**Step 1**. Embedding

Given a time series sequence $X=(x_{1},x_{2},\ldots ,x_{N})$ of length N, select a window length L (typically L≤N/2), and construct the trajectory matrix T. Here, K = N − L + 1, and the matrix T is an L×K Hankel matrix.

**Step 2**. Singular Value Decomposition (SVD)

The trajectory matrix T is subjected to SVD $T=U\Sigma V^{T}$. The trajectory matrix can be represented as a sum of elementary components and$$T=\sum_{i=1}^{L}T_{i}=\sum_{i=1}^{L}\sigma_{i}u_{i}v_{i}^{T},$$
where $u_{i}$ and $v_{i}$ are the i-th columns of U and V, respectively.

**Step 3**: Grouping

Based on the magnitude of the singular values, select the first r components (typically r≪L , and discard the remaining components (noise):$$T_{\mathrm{reconstructed}}=\sum_{i=1}^{r}T_{i}.$$

For example, in the analysis of brain electrical signals, the first three components may correspond to the main neural oscillation modes, while higher-order components i≥4 may represent noise.

**Step 4**. Reconstruction

The reconstructed matrix $T_{\mathrm{reconstructed}}$ after dimensionality reduction is transformed back into a time series through diagonal averaging. The formula for this transformation is given by:$${\tilde{x}}_{n}=\frac{1}{w_{n}}\sum_{(l,k)\in D_{n}}T_{l,k},$$
where $D_{n}=\{(l,k)|l+k-1=n\}$ represents the set of elements on the n-th diagonal line, and $w_{n}$ is the number of elements on the n-th diagonal line. Finally, the denoised signal $\tilde{X}=({\tilde{x}}_{1},{\tilde{x}}_{2},\ldots ,{\tilde{x}}_{N})$ is obtained.

From the perspective of information theory, this process is essentially a re-distribution and filtering of the amount of information in time series [34]. Since singular values are discrete and larger singular values correspond to modes with greater energy contributions within the system, the information content of the system can be quantified as:$$H=-\sum_{i=1}^{n}p_{i}\log p_{i},$$
where $p_{i}$ represents the proportion of the total energy occupied by the *i*-th singular value, i.e.,$$p_{i}=\frac{\sigma_{i}^{2}}{\sum_{j=1}^{r}\sigma_{j}^{2}},$$
where $\sigma_{i}$ is the *i*-th singular value, and *r* is the rank of the trajectory matrix. This entropy value reflects the energy distribution of the signal in various modes. Therefore, the information-theoretic basis of SSA denoising lies in: retaining high $p_{i}$ modes (low entropy, corresponding to signals), and discarding low $p_{i}$ modes (high entropy, corresponding to noise), thus achieving signal-noise separation.

#### 2.2.3. Feature Extraction Process Based on SSA-EMS Algorithm

The process of the EMS algorithm can be divided into two stages: spatial filter learning and data projection [19]. This paper combines the respective advantages of SSA and EMS, embedding SSA into the learning process of the EMS spatial filter, and proposes a SSA-EMS feature extraction algorithm based on a three-stage entropy control framework of “noise entropy suppression → mutual information optimization → entropy rate convergence”. The algorithm flow is as follows:**1.** **Spatial Filter Learning**

Let the dataset be denoted as X∈ℝN×C×T, where Xi∈ℝC×T represents the EEG of the *i*-th epoch (with *N* being the number of epochs, *C* the number of channels, and *T* the number of time points), and yi∈{0,1} is the binary classification label corresponding to Xi∈ℝC×T.

**Step 1**. Calculate the mean values of categories: Calculate the average response corresponding to two categories in the dataset.$${\bar{X}}_{0}=\frac{1}{N_{0}}\sum_{i:y_{i}=0}X_{i},\;{\bar{X}}_{1}=\frac{1}{N_{1}}\sum_{i:y_{i}=1}X_{i},$$
where $N_{0}$ and $N_{1}$ are the number of epochs for category 0 and category 1, respectively.

**Step 2**. Apply SSA preprocessing: Perform singular spectrum analysis on the mean values of the two categories separately to remove noise and retain important components.$${\tilde{X}}_{0}=\mathrm{SSA}({\bar{X}}_{0}),\;{\tilde{X}}_{1}=\mathrm{SSA}({\bar{X}}_{1})$$

**Step 3**. Construct the spatial filter: Calculate the differential spatial filter W=X˜0−X˜1, where W∈ℝC×T is the spatial filter matrix.

**Step 4**. Normalization processing: Normalize the spatial filter by time points column-wise.$$W_{:,t}=\frac{W_{:,t}}{\|W_{:,t}{\|}_{2}+\epsilon },\;t=1,2,\ldots ,T,$$
where $\epsilon =10^{-10}$ is a small constant to prevent division by zero.

In SSA, a signal is typically decomposed into four components. In this study, the first three principal components were retained to achieve signal denoising and feature enhancement. These first three components generally correspond to the dominant trend and prominent oscillatory patterns in the signal, encompassing key neural rhythms related to the task, while the fourth component mostly contains high-frequency noise or non-specific fluctuations [35]. Directly computing the differential spatial filter based on class-averaged signals may amplify these noise components, compromising the stability of feature extraction. Therefore, by retaining the first three components, the core neural activity features can be effectively preserved, while more robustly extracting stable differential patterns between the two conditions, thereby enhancing the reliability of the analysis.

**2.** 
**Data Projection—mutual information optimization**


In the data projection stage, the learned spatial filter W∈ℝC×T is applied to the dataset X∈ℝN×C×T, where W∈ℝC×T is the normalized spatial filter that can be calculated through Einstein summation and convention Xtransform=einsum(nct,ct→nt,X,W) to compute the projection feature vector Xtransform∈ℝN×T.

SSA-EMS employs a time-varying spatial filter W∈ℝC×T to achieve dynamic feature selection, which is essentially minimizing the conditional entropy H(Y|X), where Y represents the target emotional states. Compared with traditional static filters, this algorithm reduces the conditional uncertainty between features and emotional labels by constructing a difference space W=X^0−X^1. The optimization is achieved through:minWH(Y|Xtransform) Xtransform=〈X,W〉,
where the projected feature Xtransform is calculated using Einstein summation and dot product, optimizing the mutual information I(Y;Xtransform).

**3.** 
**Construction of Stable Representations—Entropy rate convergence**


To obtain stable neural dynamic representations and construct feature representations that can be used for classification, let Xtransform∈ℝN×T, where Xtransform(i)∈ℝT is the transformed feature for the *i*-th epoch, and y(i)∈{0,1} is its label. For each class k∈{0,1}, calculate the average feature vector corresponding to the two classes:fk¯=1Nk¯∑i:y(i)=kXtransform(i),
where Nk is the number of epochs in class *k*. This process, by averaging the same class of epochs in Xtransform, can obtain more stable neural dynamic representations.

The essence of this stage is to reduce entropy. The entropy $H_{r}(X_{transform}^{\left(i\right)})\;$ of the feature $X_{transform}^{\left(i\right)}$ within a single epoch reflects the temporal uncertainty. Under the assumption of independent and identically distributed (i.i.d.) noise, averaging over same class $f_{k}$ can lead to entropy reduction:$$H_{r}(f_{k})\leq \frac{1}{N_{k}}\sum H_{r}(X_{transform}^{\left(i\right)}).$$

This process suppresses individual random variations and enhances the generalization ability across samples.

From the above process, it can be seen that the SSA-EMS algorithm can reduce the high-dimensional feature X∈ℝN×C×T to a feature matrix F∈ℝ2×T, thus achieving the purpose of dimensionality reduction. F∈ℝ2×T is the time-domain feature vector extracted by the SSA-EMS algorithm, which can be used for subsequent classification training and recognition.

The aforementioned procedure can also employ leave-one-out or cross-validation methods for spatial filter learning and data projection to achieve better robustness. This study adopts the above procedure combined with a stratified 5-fold cross-validation method for spatial filter learning and data projection.

#### 2.2.4. Cross-Subject Data Processing in Seed Emotional EEG

This study conducts cross-subject emotion recognition research based on the first experimental EEG data of 15 subjects from the SEED dataset. Each subject’s file contains 15 EEG segments (5 positive, 5 neutral, and 5 negative). A total of 225 EEG signal samples (75 per emotional category) were collected from the 15 subjects. All EEG data were preprocessed using the MNE library into standard Raw objects [36] and segmented into 1 s epochs X∈ℝN×C×T.

**1.** 
**Cross-subject sample combination data processing**


In traditional cross-subject studies, “subject-independent” evaluation typically employs a simple strategy of splitting training and test sets, which is a coarse-grained approach that makes it difficult to precisely characterize model transfer performance under different subject combinations. To overcome this limitation, this study designs a more challenging cross-subject combination data fusion method: by computing the Cartesian product between emotional A data from each subject and emotional B data from each subject, thereby forming all pairwise combinations of samples across subjects. This method not only preserves the core advantages of “subject-independent” evaluation but also enables assessment of the model’s emotional transfer performance between subjects, suppressing interference caused by inter-subject variability. The specific implementation steps are as follows:

**Step 1**. Emotional pairing: Construct three emotional combinations: positive vs. negative, positive vs. neutral, and negative vs. neutral.

**Step 2**. Data combination: For each emotional combination (A vs. B), combine 75 samples of type A and 75 samples of type B using the Cartesian product, resulting in 5625 sample units.

**Step 3**. Data Concatenation: Concatenate the two types of emotions in each sample unit along the first dimension to form a three-dimensional matrix X∈ℝ(M+N)×C×T.

**Step 4**. Dimensionality Reduction: Apply the SSA-EMS method to reduce the three-dimensional matrix X∈ℝ(M+N)×C×T to a feature matrix F∈ℝ2×T, where each row represents the features of one emotion category.

**Step 5**. Feature Integration: After processing all 5625 sample units, construct a two-dimensional feature matrix F∈ℝ11250×T containing 11,250 (5625 × 2) feature vectors.

The specific steps are shown in Figure 1.

**2.** 
**Subject-Independent Evaluation Method**


This study also adopts the conventional subject-independent evaluation paradigm to verify the performance of the SSA-EMS algorithm. The specific implementation is as follows:

1. Data Partitioning:

Training set: EEG data from the first 7 subjects.

Test set: EEG data from the last 8 subjects.

2. Sample Construction Strategy:

For each pair of emotion combinations (A vs. B), the following procedure is adopted:

**Step 1**. For each subject, 15 samples of class A and 15 samples of class B are selected from their three experimental sessions (as noted, each subject participated in three sessions approximately one week apart).

**Step 2**. A Cartesian product operation is performed to generate 225 (15 × 15) sample pairs, all derived from the same subject. All sample pairs are strictly maintained as within-subject combinations, avoiding any cross-subject mixing.

3. Dataset Scale:

Total training set: 7 subjects × 225 units = 1575 sample units

Total test set: 8 subjects × 225 units = 1800 sample units

This scheme strictly adheres to the core requirements of subject-independent evaluation: (1) complete subject-level separation between training and test sets; (2) all data combinations maintain within-subject consistency; (3) ensuring that the model never accesses data from any test subject during training [37]. The specific steps are shown in Figure 2.

## 3. Results

### 3.1. Cross-Subject Sample Combination Classification

This experiment employs stratified 5-fold cross-validation to separately evaluate the influence of different frequency bands on the classification of emotional states for the three experimental groups: “positive vs. neutral”, “positive vs. negative”, and “neutral vs. negative”. A Random Forest classifier (n_estimators = 100) and a Radial Basis Function Support Vector Machine (RBF-SVM, kernel = ‘rbf’, C = 1, gamma = ‘scale’) are used to compare classification performance. The final experimental result is the average recognition rate from the stratified 5-fold cross-validation.

Figure 3 shows the recognition accuracy for pairwise classification of three emotional states (positive vs. neutral, negative vs. neutral, positive vs. negative) across different frequency bands (All band, Delta, Theta, Alpha, Beta, and Gamma). The x-axis represents different frequency bands, and the y-axis represents the recognition accuracy.

As shown in Figure 3, the RF outperformed SVM across all tasks and frequency bands. For instance, in the negative vs. neutral emotion classification task using the All Bands, RF achieved an accuracy of 98.61%, compared to 94.10% for SVM; in the Delta band for the negative vs. positive emotion classification task, RF achieved an accuracy of 95.11%, also higher than SVM’s 92.16%.

In the recognition of the three emotion state combinations, the performance using the full frequency band was superior to that of each individual frequency band. Notably, RF classifier achieved particularly high accuracy in the full frequency band, with classification accuracies of 98.61%, 98.44%, and 98.63% for the negative vs. neutral, negative vs. positive, and neutral vs. positive tasks, respectively.

In the single-band classification tasks, the Delta band consistently achieved the highest classification accuracy among the five individual bands (95.27% for negative vs. neutral, 95.11% for negative vs. positive, and 94.98% for neutral vs. positive). The Gamma band ranked second, with accuracies exceeding 90% across all tasks, demonstrating strong classification performance. The Theta, Alpha, and Beta bands generally yielded lower classification accuracies compared to the Delta and Gamma bands.

The confusion matrix is a fundamental tool for evaluating classification model performance, providing an intuitive visualization of the model’s discriminative ability across different classes [38,39]. Since RF demonstrated the best classification performance, Figure 4 presents the confusion matrices of the RF model on the test set using All-band features for the three binary emotion classification combinations (negative vs. neutral, negative vs. positive, neutral vs. positive), with the corresponding classification accuracies annotated. Overall, the model exhibited excellent performance across all three classification tasks, with accuracies exceeding 98%. The highest accuracy of 98.63% was achieved for the “neutral vs. positive” combination.

As evident from the confusion matrices, the diagonal elements (5544 vs. 5550, 5533 vs. 5542, 5547 vs. 5549) exhibit deep blue high-value regions, with proportions exceeding 98%, indicating the model’s exceptionally high accuracy in correctly classifying samples. The off-diagonal regions show lighter colors, corresponding to a small number of misclassified samples, specifically manifested as low false positive (FP) and false negative (FN) counts, with the values being close (81 vs. 75, 92 vs. 83, 78 vs. 76), suggesting the model does not exhibit a significant bias towards any particular class, demonstrating well-balanced classification performance.

### 3.2. Classification Results of “Subject-Independent” Evaluation

This experiment employed a Random Forest model (n_estimators = 100) to evaluate EEG-based emotion recognition in a subject-independent paradigm using a binary classification task. The final result is the average recognition rate obtained through stratified 5-fold cross-validation. The confusion matrix analysis results for the three test sets are shown in Figure 5. In Figure 5a (Neutral vs. Positive) and Figure 5b (Negative vs. Neutral), both false positive (FP) and false negative (FN) counts remain low and are similar in magnitude, indicating that the model exhibits no significant bias toward either class, demonstrating well-balanced classification performance. Notably, in Figure 5c (Negative vs. Positive), the misclassification rate for positive emotions is significantly higher than that for negative emotions (FN = 116 vs. FP = 18), indicating that positive emotion samples are more prone to misclassification.

Despite these differences, all three emotion binary classification tasks achieved excellent performance on the test set (accuracy > 96%), and comprehensive analysis indicates that the SSA-EMS algorithm possesses strong cross-subject generalization capability in cross-subject experiments.

## 4. Discussion

This study employed the SSA-EMS algorithm to process and analyze EEG data from three emotional unit combinations (neutral vs. positive, negative vs. neutral, negative vs. positive). The results demonstrate that the SSA-EMS algorithm exhibits excellent classification performance both in cross-subject data processing and in the “subject-independent” evaluation framework. This performance advantage stems from the complementary effects of SSA’s denoising capability and EMS’s strategy of maximizing inter-class differences. The physiological differences between emotional units are the fundamental cause of signal amplitude differentiation, while the mathematical principle of EMS lies in capturing the EEG pattern differences between two emotional states through spatial filters, thereby maximizing the divergence between the two classes. When brain activity differences are prominent, this also leads to larger mean differences between classes. The synergistic effect of SSA and EMS constitutes the key to the algorithm’s superior classification performance. Specifically, SSA enhances the signal-to-noise ratio by decomposing and reconstructing the trajectory matrix to remove high-frequency noise, which in turn enables the spatial filter weights computed by the subsequent EMS algorithm to achieve higher discriminability. Furthermore, the normalization mechanism of the filter amplifies the impact of intrinsic differences, essentially enhancing features by optimizing weight distribution across channels. The combined effect of these mechanisms enables the SSA-EMS algorithm to achieve significant advantages in feature extraction for emotional EEG signals.

From the perspective of information theory, the SSA-EMS algorithm constructs a three-stage entropy control framework: “noise entropy suppression → mutual information optimization → entropy rate convergence.”

1. SSA Layer: Reduces the noise entropy $H_{\mathrm{noise}}$ of the input signal and improves the signal-to-noise ratio (SNR):$$\mathrm{SNR}=\frac{H_{\mathrm{signal}}}{H_{\mathrm{noise}}}.$$

2. EMS Layer: Maximizes mutual information using a differential filter:$$I(W_{i};Y).$$

3. Stable Representation Layer: Achieves entropy rate convergence:$$H_{r}(f_{k})\to \mathrm{const}.$$

Figure 6 displays the average feature signals of three randomly selected emotional unit combinations from the full-frequency band, processed by the SSA-EMS algorithm. As shown, the average features of different emotional unit combinations exhibit significant distinguishability. It should be noted that the feature curve shapes presented in Figure 6 represent only one example among many possible configurations. Although diverse curve morphologies can be generated by different emotional unit combinations (not all shown due to space limitations), these features consistently maintain stable distinguishability. The results indicate that the SSA-EMS algorithm can effectively capture the intrinsic differential characteristics of distinct EEG emotional states, demonstrating significant universality in binary emotional classification.

This study is compared with the research conducted by Jie Quan, Yihang Dong, Longbin Jin, and their colleagues. Jie Quan et al. proposed a cross-subject emotion EEG classification algorithm based on multi-source domain selection and sub-domain adaptation. Experiments on two public datasets, SEED and SEED-IV, demonstrated that the proposed method achieved average classification accuracies of 92.83% and 79.30%, respectively [40]. Yihang Dong et al. introduced a multimodal cross-scale fusion model named “Mood Reader”. The related experiments were carried out on the SEED and SEED-V public datasets, showing that when the Mood Reader model combined DE features and employed a sliding window sequential combination (w SWC) strategy, the recognition rate was 93.12% ± 4.75% [41]. Longbin Jin and Eun Yi Kim innovatively proposed a cross-subject emotion prediction model integrating temporal analysis with spatial correlation characteristics. Experiments on the DEAP physiological signal emotion dataset and the Shanghai Jiao Tong University SEED dataset indicated significant advantages when channel features were used in conjunction with LSTM: achieving accuracies of 98.93% and 99.10% in the DEAP valence and arousal binary classification tasks, respectively, and reaching a recognition accuracy of 99.63% in the SEED three-classification task, surpassing the current state-of-the-art levels [42].

Compared with previous studies, the present research still has the following limitations: The SSA-EMS algorithm is restricted to binary classification tasks, which prevents it from handling multi-class problems simultaneously. Since emotion recognition typically involves multiple states (e.g., happiness, anger, sadness, surprise), the model cannot be directly applied to multi-class scenarios. Although three emotion categories are included in the study, they require multiple binary classifications to be combined, thereby reducing the model’s capability in distinguishing complex emotional states. Moreover, the performance of the method is also dependent on the number of available data samples, with larger datasets generally leading to improved results [43]. Therefore, extending the binary classification algorithm to a multi-class framework constitutes a direction for future research. Additionally, integrating the EEG features processed by SSA-EMS with other physiological signals to construct a multimodal fusion approach for emotion recognition could be explored.

## 5. Conclusions

The SSA-EMS algorithm proposed in this study significantly enhances cross-subject EEG-based emotion recognition performance by integrating the noise-reduction capability of singular spectrum analysis with the dynamic difference enhancement property of effect-matching spatial filtering. Experiments demonstrate that the algorithm achieves over 98% accuracy in binary classification tasks of positive, neutral, and negative emotions using All-band features, with high recognition rates also observed in the Delta and Gamma bands, and the Random Forest classifier outperforms the Support Vector Machine. Rigorous “subject-independent” evaluation further validates the model’s generalization ability on unseen subjects (accuracy > 96%), indicating that the SSA-EMS algorithm can stably extract dynamic features of emotional EEG. However, the algorithm supports only binary classification tasks, limiting its application in complex multi-emotion scenarios; future research should explore extensions to multi-class classification and strategies for multimodal physiological signal fusion.

## Figures and Tables

**Figure 1 entropy-27-00986-f001:**
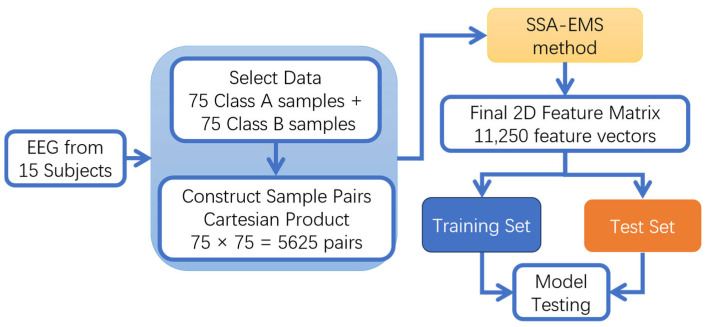
Cross-subject sample combination data processing.

**Figure 2 entropy-27-00986-f002:**
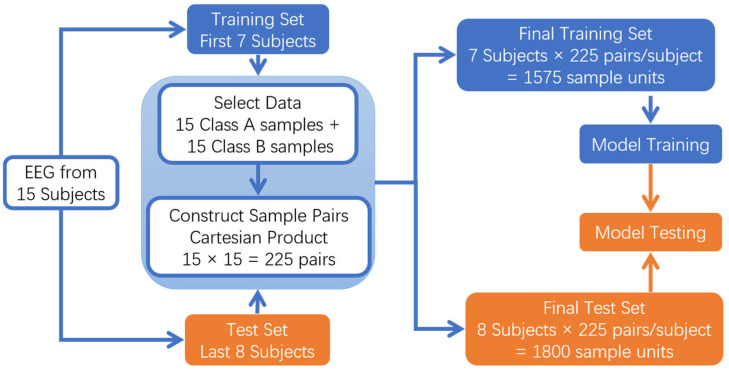
Subject-Independent Evaluation Method.

**Figure 3 entropy-27-00986-f003:**
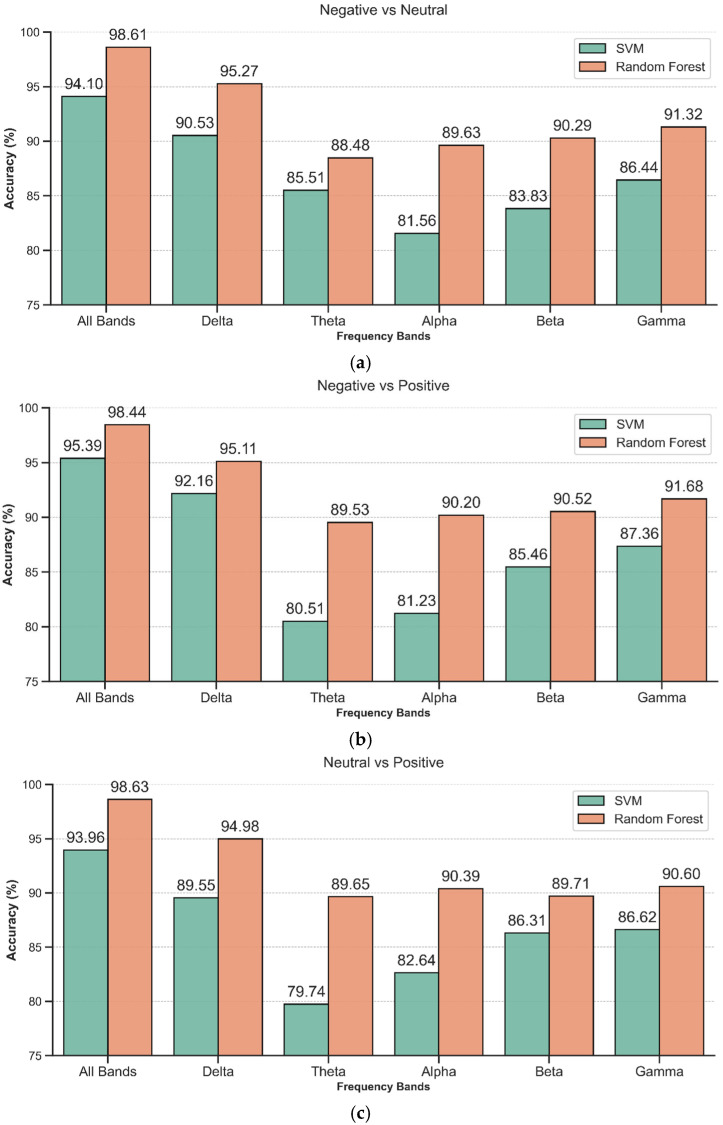
Recognition accuracy of emotional states across different frequency bands. (**a**) negative vs. neutral; (**b**) negative vs. positive; (**c**) neutral vs. positive.

**Figure 4 entropy-27-00986-f004:**
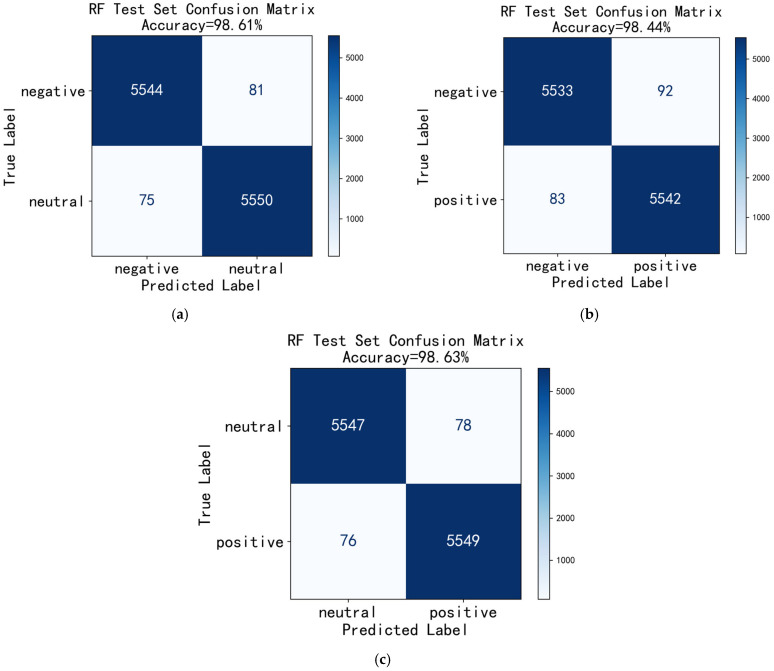
Confusion matrix of RF under All-band feature classification. (**a**) negative vs. neutral; (**b**) negative vs. positive; (**c**) neutral vs. positive.

**Figure 5 entropy-27-00986-f005:**
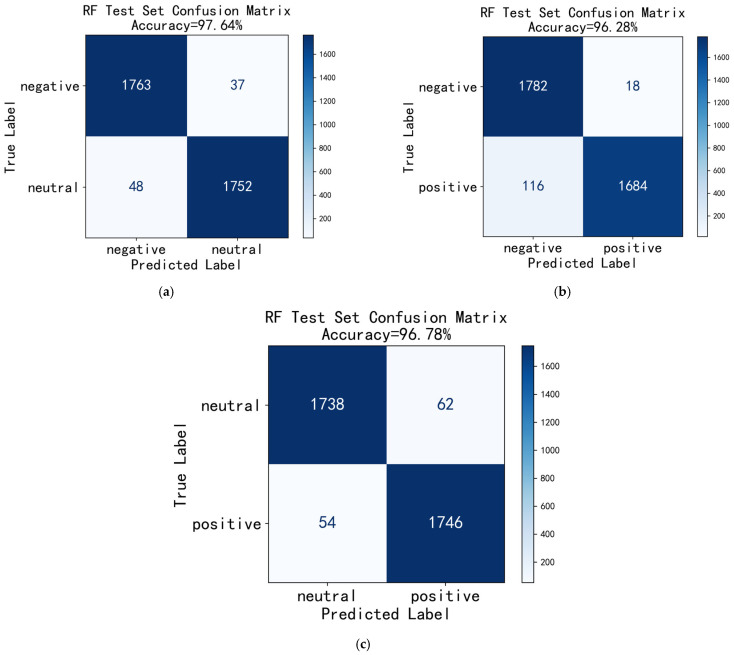
Confusion matrix of RF under “subject-independent” evaluation. (**a**) negative vs. neutral; (**b**) negative vs. positive; (**c**) neutral vs. positive.

**Figure 6 entropy-27-00986-f006:**
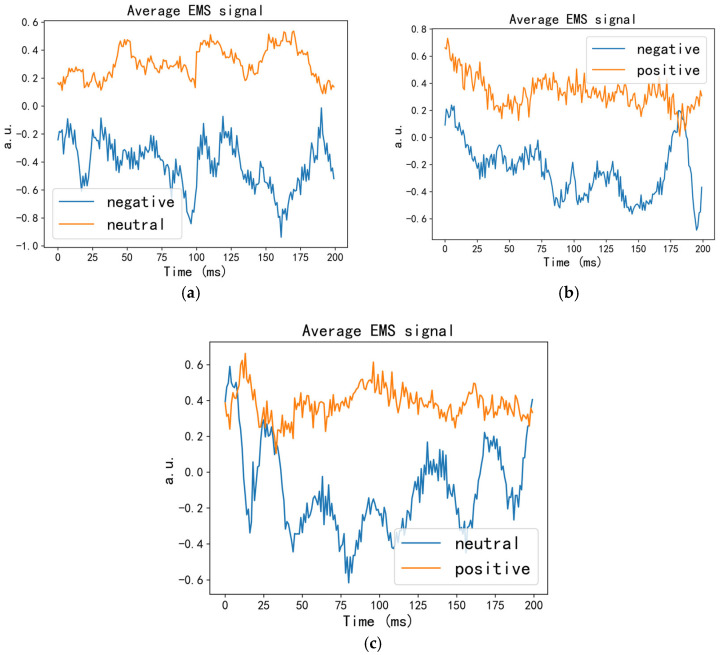
Examples of average feature signals for three emotional unit combinations. (**a**) negative vs. neutral; (**b**) negative vs. positive; (**c**) neutral vs. positive.

## Data Availability

Further inquiries can be directed to the corresponding authors.

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
