# Peer review of "Cross-Subject EEG Emotion Recognition Using SSA-EMS Algorithm for Feature Extraction"

_entropy, 2025, doi:10.3390/e27090986_

Round 1

Reviewer 1 Report

Comments and Suggestions for Authors

This work introduces an approach that combines singular spectrum analysis with effect-matched spatial filtering to improve classification accuracy in cross-subject analysis. In my opinion, the work aligns well with the aims of Entropy journal; however, several aspects require further attention. Therefore, my recommendation is major revision. Below is a detailed list of points that need clarification or improvement:

  1. In the Introduction, the acronym EEG should be defined.

  2. Introduction, lines 31–33: I am not sure about the statement that EEG is typically non-stationary. Indeed, much of the literature applies methods based on the stationarity assumption. However, it is true that in some instances, such as pathophysiological states (e.g., seizures or ERP analysis), EEG can be non-stationary. I suggest rephrasing this sentence for accuracy.

  3. Line 56: What is SEED? It is mentioned without prior introduction.

  4. Line 88: Since the authors did not perform any preprocessing steps, this subparagraph seems unnecessary and can be removed.

  5. Lines 93 and 117: Provide the full form of the acronyms used.

  6. Line 130: How was the trajectory matrix T constructed?

  7. Line 131: What is a Hankel matrix? Please provide a brief explanation.

  8. Line 133: Define the matrices U, Σ, and V.

  9. Line 135: What is L? Is it the length of the selected window or something else? Also, define σ_i.

  10. Line 139: The sentence “based on the magnitude of outliers” is unclear. Please clarify what the authors mean.

  11. Line 154: This is the definition of entropy for discrete random variables. Which random variables are considered here, and why are they treated as discrete? Please provide an explanation.

  12. Line 155: p_i should represent the probability associated with the i-th event. How does this relate to the proportion of total energy? Please clarify.

  13. Line 161: The authors state that high entropy corresponds to noise but do not explain why. Is this because noise is less predictable? Please elaborate.

  14. Line 206: It is unclear what Y represents in the context of the conditional entropy definition.

  15. Line 291: As a baseline, it would be helpful to show results without applying the proposed method. For example, what would be the classification performance if one simply performed frequency-domain analysis with cross-subject classification?

  16. Figure 1: Since the results are reported for different frequency bands, it is unclear from the preprocessing description how the features were divided into these bands.

  17. The method strongly depends on the number of data samples available for training. This limitation should be mentioned, and the authors should acknowledge related work addressing this issue [1][2].

  18. A visual representation, such as a flowchart, is needed to describe the proposed method. This would make the process easier to interpret and understand, as the current description is somewhat difficult to follow.

References:
[1] https://doi.org/10.1109/ACCESS.2024.3368637

Author Response

Thank you very much for taking the time to review this manuscript.

Comments 1: [In the Introduction, the acronym EEG should be defined.]

Response 1: [Thank you for pointing this out. I have added the definition.]

Comments 2:[ Introduction, lines 31–33: I am not sure about the statement that EEG is typically non-stationary. Indeed, much of the literature applies methods based on the stationarity assumption. However, it is true that in some instances, such as pathophysiological states (e.g., seizures or ERP analysis), EEG can be non-stationary. I suggest rephrasing this sentence for accuracy.]

Response 2:[ Thank you for your suggestion. We have rephrased it accordingly.]

Comments 3:[Line 56: What is SEED? It is mentioned without prior introduction.]

Response 3:[ Thank you for your suggestion. We have supplemented it.]

Comments 4:[ Line 88: Since the authors did not perform any preprocessing steps, this subparagraph seems unnecessary and can be removed.]

Response 4:[ We have removed that section, but kept the content to indicate the source of the database data used.]

Comments 5:[ Lines 93 and 117: Provide the full form of the acronyms used.]

Response 5:[ Thank you for your suggestion. We have supplemented it.]

Comments 6-9:[ Line 130: How was the trajectory matrix T constructed? Line 131: What is a Hankel matrix? Please provide a brief explanation. Line 133: Define the matrices U, Σ, and V. Line 135: What is L? Is it the length of the selected window or something else? Also, define σ_i.]

Response 6-9:[ Thank you for your suggestion. This is fundamental mathematical content, and we do not believe it requires further explanation in the paper.]

Comments 10:[ Line 139: The sentence “based on the magnitude of outliers” is unclear. Please clarify what the authors mean.]

Response 10:[ Thank you for pointing out the error. We have corrected it.]

Comments 11:[ Line 154: This is the definition of entropy for discrete random variables. Which random variables are considered here, and why are they treated as discrete? Please provide an explanation.]

Response 11:[ Thank you for your suggestion. We have provided a relevant explanation..]

Comments 12:[ Line 155: p_i should represent the probability associated with the i-th event. How does this relate to the proportion of total energy? Please clarify.]

Response 12:[ Thank you for your suggestion. We have provided a relevant explanation in the article. Information entropy can be used to measure the uncertainty of any distribution.]

Comments 13:[ Line 161: The authors state that high entropy corresponds to noise but do not explain why. Is this because noise is less predictable? Please elaborate.]

Response 13:[Yes, noise is less predictable. A high-entropy signal has a nearly uniform distribution of its values. Every possible value (or amplitude) is equally likely to occur. Noise exhibits this property.]

Comments 14:[ Line 206: It is unclear what Y represents in the context of the conditional entropy definition.]

Response 14:[ Thank you for your suggestion. We have supplemented it. ]

Comments 15:[ Line 291: As a baseline, it would be helpful to show results without applying the proposed method. For example, what would be the classification performance if one simply performed frequency-domain analysis with cross-subject classification?]

Response 15:[ Thank you for your suggestion. Since emotion analysis based on EEG signals is highly complex, the methods employed are generally sophisticated. Using simplistic approaches would almost certainly result in low recognition rates, while more complex methods could make the paper itself excessively lengthy. In the discussion section of the paper, we also compared our approach with those of other researchers.]

Comments 16:[ Figure 1: Since the results are reported for different frequency bands, it is unclear from the preprocessing description how the features were divided into these bands.]

Response 16:[ It is not about how these features are assigned to frequency bands, but rather about first dividing the EEG signals into different frequency bands and then evaluating the emotion classification performance of the SSA-EMS algorithm across these bands. ]

Comments 17:[ The method strongly depends on the number of data samples available for training. This limitation should be mentioned, and the authors should acknowledge related work addressing this issue [1][2].]

Response 17:[ We would be happy to cite this paper, but we do not find it entirely suitable. Moreover, the method is not significantly influenced by sample size, as the quantity of data depends primarily on the capacity of the dataset itself.]

Comments 18:[ A visual representation, such as a flowchart, is needed to describe the proposed method. This would make the process easier to interpret and understand, as the current description is somewhat difficult to follow.]

Response 18:[ Thank you for your suggestion. We have now added the flowchart.]

Thank you for providing many valuable opinions.

Reviewer 2 Report

Comments and Suggestions for Authors

The article has reached the acceptance level, but there are some minor issues that need to be modified:
1. The reference format is not consistent, it is recommended to revise it;
2. The font size of the horizontal and vertical axes in Figures 3 and 4 is too small to see clearly.

Author Response

Thank you very much for taking the time to review this manuscript.

Comments 1: [The reference format is not consistent, it is recommended to revise it.]

Response 1: [Thank you for pointing this out. Entropy journal has a specific citation format. We will actively communicate with the editors and revise it according to the formatting requirements of Entropy journal.]

Comments 2:[ The font size of the horizontal and vertical axes in Figures 3 and 4 is too small to see clearly.]

Response 2:[ Thank you for your suggestion. We have made corresponding changes by increasing the font size in the figures.]

Thank you for providing those valuable opinions.

Round 2

Reviewer 1 Report

Comments and Suggestions for Authors

The authors state that “the method is not significantly influenced by sample size, as the quantity of data depends primarily on the capacity of the dataset itself.” However, since the proposed approach is based on machine learning techniques, it is well established that such methods are strongly affected by sample size . Where do the authors demonstrate that their method is indeed robust to this limitation? In several widely used machine learning approaches were compared, showing that classification accuracy is highly dependent on the available sample size. In the present manuscript, no evidence is provided to support the claim that the proposed method is exempt from this issue.

Author Response

Thank you very much for taking the time to review this manuscript.

Comments 1: [The authors state that “the method is not significantly influenced by sample size, as the quantity of data depends primarily on the capacity of the dataset itself.” However, since the proposed approach is based on machine learning techniques, it is well established that such methods are strongly affected by sample size . Where do the authors demonstrate that their method is indeed robust to this limitation? In several widely used machine learning approaches were compared, showing that classification accuracy is highly dependent on the available sample size. In the present manuscript, no evidence is provided to support the claim that the proposed method is exempt from this issue.]

Response 1: [Thank you for your suggestions. I have revised accordingly and cited the paper you recommended.]

Thank you for providing the valuable opinions.

Round 3

Reviewer 1 Report

Comments and Suggestions for Authors

I have no further comments. It would be great for the authors to explore the effects of different paramters on the performance of the proposed approach.